# Treatment of a Large Tibial Non-Union Bone Defect in a Cat Using Xenograft with Canine-Derived Cancellous Bone, Demineralized Bone Matrix, and Autograft

**DOI:** 10.3390/ani14050690

**Published:** 2024-02-22

**Authors:** Keun-Yung Kim, Minha Oh, Minkyung Kim

**Affiliations:** 1Fatima Animal Medical Center, Daegu 41216, Republic of Korea; 2Veteregen, Hanam 12930, Republic of Korea; 3Keunmaum Animal Medical Center, Busan 48096, Republic of Korea

**Keywords:** cats, demineralized bone matrix, large bone defect, non-union, osteomyelitis, xenograft

## Abstract

**Simple Summary:**

In feline tibia fractures, limited tissue coverage may lead to complications such as open fractures and delayed healing. Non-union, attributed to factors such as poor blood supply, malnutrition, bone misalignment, infections, and bone damage, necessitates bone grafting for repair. While autografts are preferred, they may be insufficient for extensive defects, prompting the use of allografts and xenografts created through xenotransplantation processes such as deproteinized bovine bone. Past graft options were limited, but contemporary alternatives encompass a broader range, incorporating xenografts, demineralized bone matrix (DBM), and calcium-based materials. When combined with growth factors or cells, these materials significantly enhance their effectiveness. This case describes the treatment of a cat’s tibia defect using an autologous cancellous bone graft, recombinant human bone morphogenetic protein-2, and a xenograft derived from canines mixed with DBM. Despite the initial use of external skeletal fixation and antibiotic therapy for infection, a significant defect persisted even after bone healing was complete, necessitating a multifaceted approach involving additional plate placement, autograft, and xenograft. As a result, the long-term prognosis was favorable with no complications observed in the subject’s bone healing.

**Abstract:**

A 17-month-old domestic short-hair cat was referred due to a non-union in the left tibia. The initial repair, conducted 3 months prior at another animal hospital, involved an intramedullary (IM) pin and wire to address a comminuted fracture. Unfortunately, the wire knot caused a skin tract, resulting in osteomyelitis. Although the wire knot was removed at that hospital, the draining tract persisted, continuously discharging exudate. Upon evaluation, the first surgery was reassessed and revised, involving the removal of the IM pin and the application of external skeletal fixation alongside an antibiotic susceptibility test. After 118 days post-revision surgery, while some cortical continuity was observed, a significant bone defect persisted, posing a substantial risk of refracture should the implant be removed. A second revision surgery was performed, utilizing a bone plate combined with cancellous bone autograft, recombinant human bone morphogenetic protein-2, and xenograft featuring a canine-derived cancellous chip mixed with demineralized bone matrix. Remarkably, the bone completed its healing within 105 days following the subsequent surgery. Radiography demonstrated successful management of the large bone defect up to the 2-year postoperative check-up. During telephone follow-ups for 3.5 years after surgery, no complications were identified, and the subject maintained a favorable gait.

## 1. Introduction

The tibia ranks as the most frequent fracture site in cats, comprising 18.0% of all fractures, and notably contributing to 61.1% of reported non-union cases [1]. The limited soft tissue coverage along the tibia’s medial surface increases the risk of complications, including open fractures, periosteal stripping, and delayed bone revascularization during the healing phase [2,3,4].

The etiology of non-union may involve multiple factors. A poor blood supply to the affected area, coupled with a suboptimal nutritional status, can predispose to non-union fractures [5,6]. Poor apposition of the fractured bone ends, pathological fractures, large quantities of necrotic bone, or infections have also been reported as potential etiological factors [6]. Bone grafts play a crucial role in improving healing in delayed unions, non-unions, osteotomies, arthrodesis, and multi-fragmentary fractures, as well as in replacing bony loss [7,8,9,10]. While autografts traditionally prove effective in repairing defects caused by comminuted fractures and non-unions, offering necessary mechanical stability, their application becomes challenging when dealing with extensive defects due to the limited availability of bone [11,12,13]. This limitation encourages exploring the use of allografts and xenografts as substitutes [14].

Xenotransplantation, involving the intentional transfer of living cells, tissues, or organs from one species to another, results in the creation of xenografts [15]. The most extensively researched and commonly employed xenograft material is deproteinized and heat-treated bovine bone [16,17,18,19,20]. Recently, canine-derived allografts with demineralized bone matrix (DBM) products have been developed, and related research on their application has been published [21,22,23]. However, comprehensive studies on the healing of bony defects in cats using canine-derived xenografts have not been fully evaluated. This case report aims to present the healing of a feline tibia with large defects after revision surgery using autologous cancellous bone grafts, recombinant human bone morphogenetic protein (BMP)-2 (rh-BMP2), and xenografts with canine-derived cancellous chips mixed with DBM.

## 2. Case Presentation

### 2.1. History and Clinical Examination

A 17-month-old, 4.91 kg, castrated male domestic short-haired cat presented with a 3-month history of non-weight-bearing lameness in the left hindlimb. The subject had sustained a fracture in an accident 3 months prior and had subsequently undergone treatment involving intramedullary (IM) pinning and wiring at a different hospital. However, complications arose as the knot on the wire became abscessed, necessitating its removal, leaving only the IM pin in place. Upon arrival at our facility, the left tibia exhibited palpable crepitus, and evident signs of inflammation, including edema, redness, warmth, and pain, were observed. Additionally, a suppurative exudate was evident through the skin tract (Figure 1A). However, the patient had not received any prescribed medications, including antibiotics, at the previous hospital, with only chlorhexidine being used for skin disinfection.

### 2.2. Anesthesia and Surgical Treatment

The subject’s pre-anesthesia examinations, including blood work and thoracic radiography before surgery, were unremarkable. Before IM pin removal, a sample for antibiotic susceptibility testing was taken from the skin tract, and the infection site was thoroughly irrigated with normal saline until it appeared pale. Preoperative medication included prophylactic cefazolin (25 mg/kg IV; Cefazolin injection 1 g, Chongkundang, Seoul, Republic of Korea) and premedication with acepromazine (0.03 mg/kg IV; Sedaject injection, Samu median, Yesan, Republic of Korea), and ketamine (5 mg/kg IV; Ketamine HCl injection, Huons, Seongnam, Republic of Korea). General anesthesia induction employed propofol (6 mg/kg IV; Provive injection 1%, Phambio Korea, Seoul, Republic of Korea), and maintenance was carried out with isoflurane (Ifran liq., Hana Pharm, Seoul, Republic of Korea) inhalation anesthesia at an oxygen flow rate of 2 L/min. The proximal part of the IM pin was observed to protrude into the stifle joint and was therefore excised aseptically through a stab incision, employing a combination of chlorhexidine gluconate and alcohol for skin asepsis. Subsequent radiographic examination revealed a non-union in the mid-diaphysis of the left tibia (Figure 1B,C). For external skeletal fixation, the subject, positioned in left recumbency, underwent aseptic surgery. Utilizing fluoroscopy, external skeletal fixation was placed using a 2.0 mm positive threaded pin. The proximal tibial fragment was stabilized in a type 1a configuration, while the distal fragment was secured in a type 1b configuration. The pin ends were appropriately bent to align parallel to each other and then secured with wire before being further reinforced with epoxy putty (Figure 2A).

### 2.3. Postoperative Management and Prognosis

After the surgery, the subject remained in the hospital for 2 weeks. By the third postoperative day, the subject initiated small steps, progressing to a gradual improvement in gait, eventually resulting in discharge from the hospital with restored normal ambulation. Following the revision surgery, the patient received cefazolin at a dosage of 25 mg/kg intravenously twice a day (BID), famotidine at 0.5 mg/kg intravenously BID, tramadol at 2 mg/kg intravenously BID, and meloxicam at 0.3 mg/kg subcutaneously as a single shot, along with once-daily dressing changes. On the fifth day post-revision surgery, antibiotic susceptibility testing revealed cephalosporin resistance, prompting a change in antibiotic therapy. The antibiotic susceptibility test confirmed *Enterococcus* spp., resistant to various antibiotics, but susceptible to amoxicillin–clavulanate (AMC). Consequently, the subject received a 1-month course of AMC (12.5 mg/kg PO, twice a day; Amocla Tab. 375 mg, Kuhnil, Seoul, Republic of Korea) with a gastric protectant in addition to the treatment.

The exudation from the tibia ceased a week after the revision surgery, and the infection was successfully resolved with complete wound healing observed at 10 days after the revision surgery; however, the bone gradually became more osteolytic without forming a callus (Figure 2B–D). Ultimately, while some cortical continuity was observed in the caudo-lateral aspect of the tibia, a substantial defect remained in the cranio-medial aspect of the tibia, and the potential for re-fracture was deemed significantly high if the external fixator were to be removed. Consequently, a decision was made, following the evaluation of 118-day post-operative radiographs, to remove the external fixator and address the defect by filling it with bone substitutes, which was accompanied by additional fixation utilizing a plate.

The patient was anesthetized and prepped in the same manner as for the initial revision surgery. During the medial approach of the tibia, a closed IM cavity was observed (Figure 3A). The inactive bone margin was debrided using a burr and rongeur, which was followed by rimming the bone marrow cavity with a drill bit to ensure better visibility of blood flow (Figure 3B,C). All procedures were conducted with chilled saline to prevent thermal damage and aimed to minimize bone loss. Subsequently, the tibia was stabilized using a 1.5/2.0 mm locking compression plate (DePuy Synthes Vet, Solothurn, Switzerland) positioned on the medial surface and secured with 2.0 mm locking screws. Three screws were inserted into the proximal bone fragment, and two screws were placed in the distal bone fragment. Following the application of plates and screws, autologous cancellous bone was aseptically harvested from the left humeral head. This bone was then combined with synthetic substitutes, cancellous bone derived from canines mixed with DBM (NatraOss; Veteregen, Hanam, Republic of Korea), along with rh-BMP2 (Novosis; CGBIO, Seoul, Republic of Korea). The bone graft mixture was meticulously applied to fill the large defect beneath the plate, and the skin was closed routinely (Figure 3D). The subject exhibited good mobility both before and after the surgery, prompting their discharge home on the third day post-operation.

At 28 days after the second revision surgery, the radiographic examination revealed cortical bridging, and the bone graft exhibited signs of resorption, although some areas remained unbridged (Figure 4B). However, by the 105-day mark post-surgery, a subsequent check confirmed complete cortical bridging across all areas, which was accompanied by bone remodeling (Figure 4C). At the 2-year postoperative follow-up, the subject exhibited an uneventful physical exam, a normal gait, and radiographic evidence indicating further cortical bone remodeling and reshaping compared to the previous assessment at the graft site (Figure 4D). Subsequently, phone follow-ups regarding the subject were undertaken until 3.5 years post-surgery, confirming normal walking ability and the absence of any complications.

## 3. Discussion

Fracture healing necessitates osteogenic cells, an osteoconductive matrix, an osteoinductive stimulus, mechanical stability, and adequate vascular supply [24]. Deficiency in any of these elements can lead to delayed union or non-union, with non-union fractures developing under various circumstances, including inadequate fixation, compromised blood supply, infection, presence of soft tissue between fragments, or substantial gaps between fracture fragments [1,25,26]. In our subject’s case, osteomyelitis and insufficient mechanical stability due to inadequate implant selection (IM pin and wire) led to non-union at presentation. Additionally, the distal location of the fracture in the tibia, commonly associated with longer healing durations compared to proximal long bones, could be attributed to diminished soft tissue coverage and a relatively poorer blood supply [1]. To address these issues, we implemented external skeletal fixation and antibiotic therapy to treat the infection, which was followed by the restoration of the bone defect using a xenograft along with bone plate application.

External skeletal fixation was chosen as the primary method to address the infection due to concerns that plates might exacerbate it by extending the affected area. Additionally, suspected low bone density resulting from the infection and lack of use necessitated the need for bone stabilization. The external skeletal fixator can be removed once clinical union, defined as the presence of a bridging callus on three of four cortices on two orthogonal radiographic views, has been reached [27]. However, in our case, even though the surgical site infection was effectively treated and confirmed partial bone healing following the initial revision surgery, the substantial defect within the bone created a high likelihood of bone re-fracture if the implant was to be removed. Hence, our strategy entailed addressing the bone defect by employing bone graft material and reinforcing stabilization with an additional plate fixation.

Segmental cortical defects averaging 1.5 times the bone diaphyseal diameter are known to impair bone healing in dogs and cats [26]. Enhanced bone regeneration is justified in cases of non-union not only to provide support and fill existing lacunae but also to enhance biological repair when the skeletal defect reaches the so-called critical size [28]. Over the years, the use of bone substitute materials has traditionally been limited to cancellous and cortical autografts or allografts; however, modern alternatives now encompass a range of bone graft substitutes, including xenografts, DBM, and calcium-based materials. Ongoing research explores the incorporation of growth factors, osteoprogenitor cells, or both, in combination with bone grafts or graft substitutes [29]. The final selection of which bone substitute material to use is subsequently based on the specific requirements of the actual clinical situation [30].

In our subject, the limited availability of cancellous bone autograft compared to the defect size prompted the exploration of additional methods to comprehensively address the issue. To address this clinical situation, we adopted a product combining canine-derived cancellous bony chips with DBM as a xenograft and rh-BMP2, alongside cancellous bone autograft, aiming to increase the engraftment success rate for this subject [31]. Although xenografts are convenient for storage and use, they possess considerably less capacity for osteoinduction and osteoconduction compared to autografts [14]. To address the shortcomings of these xenografts, additional materials, such as rh-BMP2 and cancellous bone autograft, have been incorporated.

The incorporation of cancellous bone xenografts combined with DBM, derived through the decalcification of cortical bone, significantly enhances osteoinductive properties. This process reduces mineral content, enhancing graft flexibility for precise placement. Moreover, combining DBM with adjuncts like hydroxyapatite, autografts, or bone marrow aspirate further refines its manipulative qualities and mechanical strength. Consequently, we integrated it with a small quantity of autograft harvested from the humeral head [20,32].

Furthermore, the integration of BMPs was employed to bolster osteoinductive capabilities. BMPs serve as differentiation factors, which are primarily responsible for prompting the transformation of undifferentiated mesenchymal cells into chondroblasts [33]. It was suggested that BMP2, BMP6, and BMP9 may play an important role in inducing the osteoblast differentiation of mesenchymal stem cells, whereas, in contrast, most BMPs are able to stimulate osteogenesis in mature osteoblasts [34]. Recent recombinant technology has allowed the isolation, production, and application of these synthesized molecules for osteoinductive and osteoconductive purposes required for healing bone defects [35]. Interspecies amino acid sequence homology for rh-BMP2 is 100% in most mammalian species, thus allowing for its use in all species that are commonly treated by veterinary fields [14].

The most commonly used xenograft is bovine-derived, and there have been several successful applications in humans, dogs, and cats [16,36,37]. Xenografts demonstrate promise; however, it is essential to take into account the possibility of immune rejection, which has been primarily attributed to the presence of a cell and matrix surface carbohydrate antigen known as the α-galactosyl epitope [38]. The presence of foreign material can trigger an antigenic response, heightening the risk of graft rejection, with a greater likelihood of rejection observed when utilizing pure bone xenografts [14]. To address these issues, the process for bovine-derived cancellous bone material involves boiling, defatting, and deproteination while aiming to preserve the stiffness, internal structure, and minimize immune responses [39].

Recent reports have documented successful applications of allografts in dogs, but using dog-derived bone grafts in cats remains undocumented [22]. Foreseeing decreased complications owing to the xenograft’s pre-processing and anticipating diminished immune reactivity in our subjects, the adoption of a canine-derived xenograft was contemplated to potentially aid the bone-healing mechanism, leading to its selection for our subject’s treatment.

This report represents the first instance, to the authors’ knowledge, of using a canine-derived xenograft to address non-union in a cat. Nevertheless, certain limitations exist within this case report. Firstly, due to its nature as a case report rather than a research study, objective comparisons with other established grafts for effectiveness were not feasible. Secondly, during the initial 2 years, the subject underwent radiographic follow-ups; however, subsequent monitoring has solely relied on phone follow-ups. While the subject’s gait remains favorable, conducting longer-term bone assessments has not been feasible, posing limitations in tracking bone status over time. Despite some limitations, this case demonstrated successful bone union and regained normal gait without complications. The use of canine-derived xenografts, similar to conventional bovine-derived xenografts, presents promising advantages in addressing significant non-union bone defects in cats, particularly when applied alongside autologous bone and various other bone graft substitutes to enhance outcomes.

## 4. Conclusions

In cases of significant bone defects resulting from non-union, exclusive reliance on autografts may be insufficient, prompting the need for alternative approaches. Our subject exhibited positive outcomes following treatment with a canine-derived xenograft, akin to the commonly used bovine-derived xenograft, complemented by autograft and rhBMP-2 to mitigate xenograft limitations [20]. This combined therapeutic regimen resulted in the subject’s successful recovery devoid of complications, indicating the potential efficacy of canine-derived xenografts as a viable treatment option for feline subjects managing substantial bone defects.

## Figures and Tables

**Figure 1 animals-14-00690-f001:**
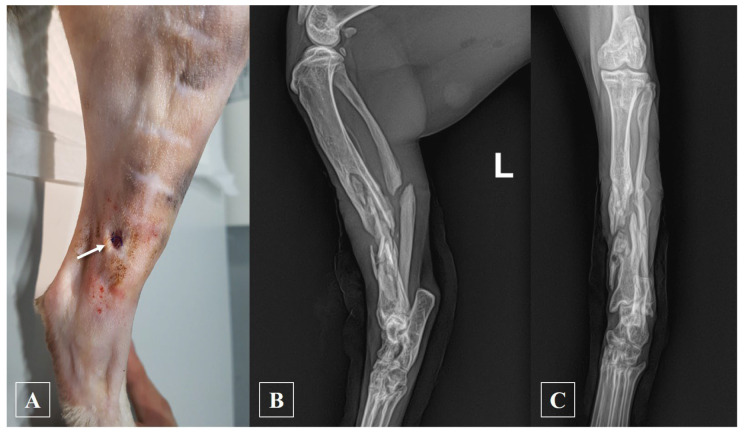
The appearance of the subject during the initial visit and subsequent radiographs following the removal of the intramedullary pin. (**A**) Following clipping, a skin tract and suture knot were observed on the medial side of the subject’s tibia, which was accompanied by exudate discharge (arrow). A sample was collected from this area and sent for antibiotic susceptibility testing. (**B**) Mediolateral and (**C**) craniocaudal radiographs of left tibia. (L: left).

**Figure 2 animals-14-00690-f002:**
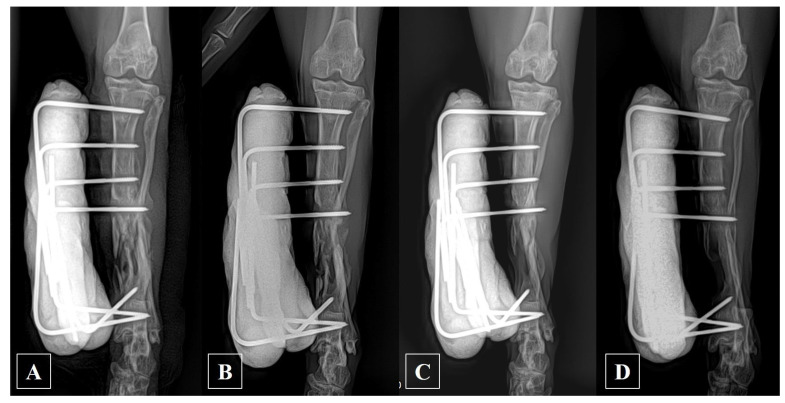
Craniocaudal postoperative radiographs of the left tibia depicting the utilization of external skeletal fixation with 2.0 mm positive thread pins and epoxy putty. (**A**) Immediately after surgery; (**B**) 14 days post-surgery; (**C**) 56 days post-surgery; and (**D**) 118 days post-surgery.

**Figure 3 animals-14-00690-f003:**
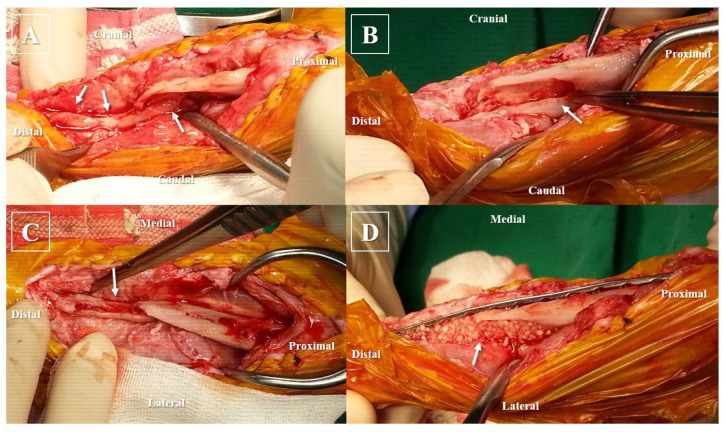
Intraoperative photos illustrating the subject’s procedure. (**A**) Immediately after soft tissue separation, significant defects and obstructed medullary cavities were revealed (arrows); (**B**) the final size of the proximal defect is shown in the lateral view (arrow). The proximal bone margin underwent decortication and was rimmed with a burr, rongeur, and drill-bit; (**C**) craniocaudal perspective displaying the distal bone after decortication, indicating visible blood supply from the bone (arrow); (**D**) application of bone substitutes to fill the defect after plate application (arrow). This included a mixture of autologous cancellous bone, recombinant human bone morphogenetic protein-2, and a canine-derived cancellous chip mixed with demineralized bone matrix.

**Figure 4 animals-14-00690-f004:**
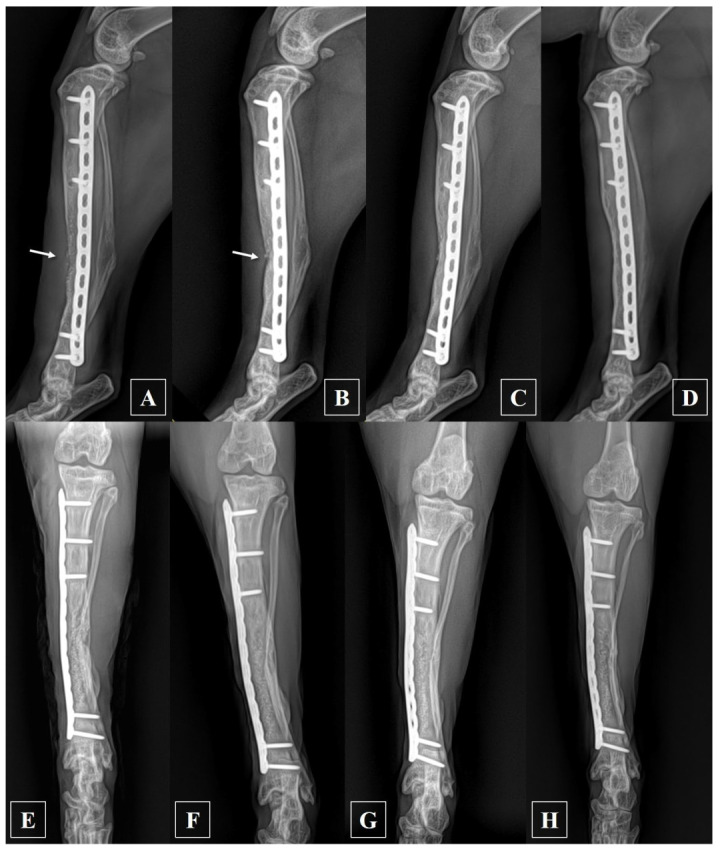
Mediolateral postoperative radiographs of the left tibia were taken with slight internal rotation to assess the plate-concealed bone area. (**A**) At 7 days after surgery, there was no bridging of the cortex observed, and the bone substitutes remained visible (arrow); (**B**) at 28 days after surgery, the bridging of the cortex was confirmed; however, there was still no continuity observed in certain areas (arrow); (**C**) at 105 days post-surgery, bridging of the cortex was observed in all sections; (**D**) two years post-surgery, further bone remodeling occurred, enhancing the distinction between cortical and cancellous bone. Craniocaudal postoperative radiographs of the left tibia: (**E**) immediately after surgery; (**F**) 42 days post-surgery; (**G**) 105 days post-surgery; (**H**) two years post-surgery.

## Data Availability

The data presented in this study are available on request from the corresponding author. The data are not publicly available due to privacy or ethical restrictions.

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
