# Peer review of "Treatment of a Large Tibial Non-Union Bone Defect in a Cat Using Xenograft with Canine-Derived Cancellous Bone, Demineralized Bone Matrix, and Autograft"

_animals, 2024, doi:10.3390/ani14050690_

Round 1

Reviewer 1 Report

Comments and Suggestions for Authors

Dear Authors, 

Thank you for contribution of this article. 

Overall this is a good article regarding the use og BMP and xenograft in a tibial fracture defect in a cat. 

The case report lacks the proper time line and the proper follow through. Time line is vague and not clear and lacks transparency. 

Abstract  summary- The timeline jumps back and forth and is inconsistent. It also does not add up. Number of surgeries and timeline should be clearly stated. Ideally in days rather than months.

Line 11 – Please change to limited tissue coverage may lead to complications

Line 28 use draining tract instead of - the skin tract 28 persisted, continuously discharging exudate

Line 31 – I thought the patient presented 3 months after the initial surgery?

This is very confusing.

Line 61 – English – please rephrase,

Line 116 – it is enough if only the antibiotic to which it is suspectable is listed, Please strike the “including cephalosporin”

Line 123 – “over time” and the bone gradually became osteolytic”  is unspecific. Please use concrete time lines here with days from second surgery.

Line 153 – At 1. Month post 3rd surgery right? Please clarify. This paper would be much better to resd if one used days form the initial surgery forward. It is hard to follow how far along we are now from the 1st surgery. No proper timeline here.

Line 189 – partial bone healing. Not continuity

Comments on the Quality of English Language

The English is of moderate quality and will need editing. 

Reviewer 2 Report

Comments and Suggestions for Authors

This case report is about the use of autologous cancellous bone grafts, recombinant human bone morphogenetic protein (BMP)-2 (rh-BMP2), and xenografts with canine-derived cancellous chips mixed with demineralized bone matrix (DBM) for the treatment of a large tibial non-union bone defect in a cat. The aim of the authors is to evaluate the short-, medium- and long-term clinical and radiographic outcome of the healing of bony defects using canine-derived xenografts, which has not been used in cat. 

The topic has a great scientific interest considering the difficulty of treating tibia non-unions in cats and treating large bone defects.

The introduction section should be improved adding more recent bibliography. 

In my opinion, the bibliography about this argument and treatments in general is poor, in particular some recently published articles are missing. Please read, add information and add the following sources to the bibliographies:

  • “Dugat D. Et al. Quantitative analysis of the intramedullary arterial supply of the feline tibia. Vet Comp Orthop Traumatol 2011; 24: 313–319 [doi: 10.3415/VCOT-11-02-0025]” to talk about the altered vascularity that can affect bone healing (L43-47);
  • Innes JF, Mint P.  Demineralised bone matrix in veterinary orthopaedics: A review. Vet Comp Orthop Traumatol 2010; 23: 393–399 [doi: 10.3415/VCOT-10-02-0022]”, which is a review about the use of DMB in veterinary orthopedics. 

The Case Presentation should be clarified and improved:

L72. Replace “3-month” with “3-months”.

L74-76. In my opinion, it’s more correct says that the implant became infected and not the single wire knot. In addition, are there the radiographic images pre-removal?

L84-85 (Figure 1). Replace “Mediolateral and (c) Craniocaudal radiograph of left tibia” with “Mediolateral and (c) craniocaudal radiographs of left tibia”.

L97. After the description of the anesthesiological protocol, the authors should describe the aseptic preparation of the patient.

L102-104. Why did authors use a type 1a in the proximal tibial fragment? A unilateral uniplanar (type Ia) construct is the least stable construct but can be used to stabilize simple fractures in immature animals, or when load sharing occurs between the bone and the fixator, as in transverse fractures. Bilateral uniplanar (type II) or unilateral biplanar (type Ib) constructs are indicated when no load sharing is possible, as in oblique/spiral fractures repaired without auxiliary fixation. Unilateral biplanar (type Ib) and bilateral biplanar (type III) constructs are used when no load sharing is possible and there is a very short proximal or distal segment, as often occurs with non reducible comminuted fractures.

L108 (Figure 2). They are not mediolateral radiographs, they are caudocranial. In addition, it is important showed to show both projections. 

L115-119. How exactly were the post-operative therapies? Describe them better. Did you start the antibiotic immediately after surgery? Was nothing given to manage inflammation and pain?

L120-121. How long after surgery has the wound healed? And how long after the application of the external fixator it was decided to remove the fixator and perform the second surgery?

L120-121. Perform a better description of the x-ray images. 

L127. In my opinion, it is the case to carry out another sub-paragraph and to describe the operation since the preparation of the patient under anesthesia.

L130. Why “chilled saline”? It is sufficient saline solution at room temperature to prevent overheating of the system. 

Figure 3. Improve image quality, especially the C. 

Figure 4. It is necessary to show also the caudocranial projections in order to be able to evaluate well the bone healing on all the cortical.

L53. Better describe the X-ray: which corticals have not yet healed for example, where healing and bone callus is observed. 

L175-177. The sentence “In our subject's case, osteomyelitis triggered by infection, coupled with insufficient mechanical stability due to a lone IM pin for a comminuted fracture, led to non-union at presentation” could be modified: “In our subject's case, osteomyelitis and insufficient mechanical stability due to inadequate implant selection (IM pin and wire), led to non-union at presentation”. In addition, it is necessary add references which indicates the correct treatment for this kind of tibial fracture. 

L180. After “[…] body supply.” add references. 

L264. After “[…] the commonly used bovine-derived xenograft.” add references.

The topic is interesting, but in my opinion it is necessary to improve the writing and implement the bibliography, especially with more recent reference. 

Reviewer 3 Report

Comments and Suggestions for Authors

It seems to me to be a very interesting work, well written and structured and with a spectacular therapeutic result, so I recommend publishing it in this form.

Round 2

Reviewer 2 Report

Comments and Suggestions for Authors

The manuscript has been improved and the bibliography implement. The authors satisfied my reviews and clarified my doubts.  

I just have one more doubt to clarify. Previous colleagues, who treated the fracture and removed the circling, set up antibiotic therapy? When the patient showed up in your clinic, it was on medical therapy? If so, which? Can you explain in the paragraph "history and clinical examination"?

As regards Figure 14, I get it that the fibula superimposed with the tibia doesn't permit an adequate assessment of the bone healing in craniocaudal view, but it is interesting to evaluate and see the other projection, and whatever orthopedic surgeon will read your manuscript will be interested to see the other projection to try to observe all four corticals, although the presence of fibula can make it difficult to evaluate. I suggest to insert the image in the manuscript. 
